# Development and Validation of the Parental Smartphone Use Management Scale (PSUMS): Parents’ Perceived Self-Efficacy with Adolescents with Attention Deficit Hyperactivity Disorder

**DOI:** 10.3390/ijerph16081423

**Published:** 2019-04-21

**Authors:** Yi-Ping Hsieh, Cheng-Fang Yen, Wen-Jiun Chou

**Affiliations:** 1Department of Social Work, College of Nursing and Professional Disciplines, University of North Dakota, Grand Forks, ND 58202, USA; yiping66@gmail.com; 2Department of Psychiatry, School of Medicine, and Graduate Institute of Medicine, College of Medicine, Kaohsiung Medical University, Kaohsiung 807, Taiwan; 3Department of Child and Adolescent Psychiatry, Chang Gung Memorial Hospital, Kaohsiung Medical Center and College of Medicine, Chang Gung University, Kaohsiung 833, Taiwan

**Keywords:** smartphone use, parental management, scale validation, attention deficit hyperactivity disorder (ADHD), smartphone addiction

## Abstract

The psychometric properties of the Parental Smartphone Use Management Scale (PSUMS) and its prospective relationships with symptoms of smartphone addiction and attention deficit hyperactivity disorder (ADHD) were studies in a sample of parents of adolescents with ADHD. This is a scale to measure parents’ perceived self-efficacy on managing their children’s smartphone use. Construct validity (exploratory factor analysis and confirmatory factor analysis), criterion-related validity (known-group validity and concurrent validity), and reliability (Cronbach’s alpha) were performed for data analyses. The results showed that the PSUMS had good factorials validity and high reliabilities, with Cronbach’s alphas ranging between 0.93 and 0.95. The 17-item PSUMS accounted for 78.58% of the total variance and contains three theoretically and statistically appropriate subscales: reactive management, proactive management, and monitoring. Strong relationships were found between parental smartphone use management and symptoms of smartphone addiction and ADHD in expected directions. Moreover, parents of children with smartphone addiction yielded lower scores on all three PSUMS subscales than parents of children without smartphone addiction. The PSUMS is considered a valuable and reliable tool in the study of parental management on their adolescent children’s smartphone use, while providing us with important targets for intervention.

## 1. Introduction

Rapid and continuing growth in the ownership of smartphones means they have now become a central gateway to online services and information. In Taiwan, approximately 73.4% of people own a smartphone, compared with 71.8% in Singapore, 70.4% in South Korea, 63.9% in the United States, and 43.8% in China [1]. In terms of age, smartphone ownership is highest among younger adults aged 18–29 (85%) [1]. The ownership of smartphones among teenagers has also surged. For instance, Adestra [2] found that approximately 87% of teenagers aged 14–18 years have smartphones, compared with 92% of adults aged 19–34 years and 65% of adults aged 56–67 years. In Taiwan, smartphone ownership among adolescents (aged 12–17 years) ranges from 78.8% to 93.3% and is 48.7% among children (aged 6–11 years) [3]. Although half (51.3%) of elementary school children do not own a smartphone, 38.8% of these children often use a family member’s smartphone. Among adolescents, smartphones are the most widely used computing devices (87%), followed by laptops (83%), tablets (51%), and desktop computers (43%) [2]. Adolescents often used a smartphone after school (82%), during class recess (74%), while taking transportation (63%), while eating food (57%), and in class (16%) [4].

Compared with adults, adolescents are more vulnerable to smartphone addiction because of the developmental stage of their brains and because they tend to have lower levels of self-control over their impulses to pursue pleasure [5,6]. A 2015 study by the Taiwan Ministry of Education reported that the prevalence of smartphone addiction among adolescents in Taiwan ranges from 13.9% to 25.7% and is even more prevalent than Internet addiction (12.6–19.5%). Smartphone addiction is associated with health problems and negative social, psychological, and behavioral effects such as sleep disturbance, depression, anxiety, low emotional intelligence, poor academic performance, risky behaviors (using a smartphone while driving or walking), and problematic behaviors [7,8,9,10]. Adolescent smartphone addiction is therefore considered a serious problem that warrants prevention and intervention.

Aside from smartphone addiction, online safety is also a major parental worry as an increasing number of children and teenagers gain access to the Internet through smartphones and tablets [11,12]. Among the common concerns that parents have about their children’s Internet and smartphone use are Internet and smartphone addiction, cyberbullying, online solicitation, sexting, cyberstalking, online friendships, online reputation, and privacy issues [12,13]. Cyberbullying and sexting, for example, are associated with psychological and behavioral health problems among adolescents [14,15] and have even been linked to suicide ideation and attempts [16,17].

Erikson’s theory of stages of development suggests that adolescents and young adults (in the fidelity and love stage) are eager to establish friendships and intimate relationships online and to blend their identities with those of their friends; they are also concerned about how they appear to others [18,19]. However, online environments may create a false sense of intimacy and intention. Cyber-stalkers may simulate ideal personas that lead adolescents to misjudge the intentions behind their messages; thereafter, cyber-stalkers may engage in obsessive relational intrusion which can be defined as “the repeated and unwanted pursuit of intimacy through violation of physical and/or symbolic privacy” [20]. An adolescent’s stage of cognitive development and their lack of life experience, along with the anonymity of the online environment, leads to greater risk-taking behavior which puts them at risk and compromises their online safety. Given the potential risks of cybercrime and problematic Internet and smartphone use, it is essential for parents to manage, educate, and communicate with their adolescent children and guide their use of digital technology.

Attention deficit hyperactivity disorder (ADHD) is the most common psychiatric disorder among adolescents with an Internet addiction [21]. ADHD has been found to be associated with Internet addiction in both cross-sectional [22] and longitudinal studies [23]. The symptoms and characteristics of ADHD include a tendency to be bored, impaired inhibition, and low achievement, and it is associated with a higher risk of Internet addiction [24] and higher levels of parent–child conflict [25]. Parents should be able to educate their adolescent children and communicate with them about online behavior and safety, as well as executing plans to manage their adolescents’ Internet and smartphone use. Therefore, a reliable metric is needed to assess self-perceived parents’ self-efficacy regarding management of their ADHD adolescent children’s smartphone use. The Parental Smartphone Use Management Scale (PSUMS) was therefore developed and validated in this study. In sum, this study tends to understand parents’ self-efficacy and engagement in managing children’s smartphone use, explore the useful parenting strategies to prevent children’s smartphone addiction and victimization, and develops a reliable and valid scale.

## 2. Materials and Methods

### 2.1. Participants and Procedure

Parents of adolescents aged between 11 and 18 years who had received a diagnosis of ADHD according to the diagnostic criteria in the Diagnostic and Statistical Manual of Mental Disorders, Fifth Edition (DSM-5) [26] were recruited for this study between August 2014 and July 2015 from the child and adolescent psychiatric outpatient clinics of two medical centers in Kaohsiung, Taiwan. Two child psychiatrists conducted diagnostic interviews with the parents and made ADHD diagnoses based on the criteria in the DSM-5. Multiple data sources—including clinical observation of each adolescent’s behavior and parental ratings of ADHD symptoms on the short version of the Swanson, Nolan, and Pelham, Version IV, Scale—Chinese version (SNAP-IV) [27,28]—were used to support each diagnosis. Adolescents who had an intellectual disability or autism spectrum disorder with difficulties in communication were excluded. Parents who had an intellectual disability, schizophrenia, bipolar disorder, or any cognitive deficits that resulted in significant communication difficulties were also excluded. A total of 237 parents of adolescents with an ADHD diagnosis were invited to participate in the study. Of these, six (2.5%) declined to participate and 20 (8.4%) reported that their children had not used smartphones in the previous month. Thus, 211 (89.0%) of the invited parents participated in the study and were interviewed by research assistants using the research questionnaire—the PSUMS, the Problematic Cellular Phone Use Questionnaire (PCPU-Q), and the short version of the SNAP-IV Chinese version. The Institutional Review Boards (IRBs) of Kaohsiung Medical University and Chang Gung Memorial Hospital, Kaohsiung Medical Center, approved the study (KMUHIRB-20130131). All participants were fully informed about the study and provided informed consent.

### 2.2. Measures

#### 2.2.1. Parental Smartphone Use Management Scale (PSUMS)

Prior to developing the PSUMS, an item pool was established by conducting a literature review and a focus group. The item pool contained 20 items. A 7-point Likert scale was used to rate the level of agreement with items, ranging from 0 (no efficacy at all) to 6 (very strong efficacy). To ensure the face validity of the scale, psychiatrists, psychologists, and parenting specialists were consulted about the content. In response to their opinions and criticism, the scale was revised to use culture-sensitive wording and remove irrelevant content. Parents were then asked to rate their self-perceived efficacy in managing their children’s smartphone use over the previous month. High PSUMS scores indicate high levels of self-efficacy.

#### 2.2.2. Problematic Cellular Phone Use Questionnaire (PCPU-Q)

The PCPU-Q was used to assess the criterion-related validity of the PSUMS. This scale measures adolescents’ problematic smartphone use [29] and comprises 12 items, each of which is rated on a 4-point Likert scale. It was developed according to the taxonomies of substance dependence classified in the Diagnostic and Statistical Manual of Mental Disorders, Fourth Edition (DSM-4) (Text Revision) [30] and modified according to the taxonomies of substance use disorder classified in the DSM-5 [26]. The PCPU-Q comprises two subscales—symptoms of problematic cellular phone use (CPU) and functional impairment caused by CPU. The first eight items address symptoms of problematic CPU among adolescents in the preceding month, including tolerance, withdrawal, CPU for longer periods of time or at higher frequencies than intended, persistent desire or unsuccessful attempts to reduce CPU, excessive time spent on CPU, giving up or reducing the number of important social, academic, or recreational activities because of CPU, continued heavy CPU despite knowledge of the physical or psychological problems caused, and an urge to use the cellular phone. The five remaining items gauge the functional impairment exhibited by an adolescent in the preceding month as a result of CPU. Higher overall scores indicate more severe levels of problematic CPU symptoms and CPU-related functional impairments. The internal reliability (Cronbach’s α) of the PCPU-Q subscales was 0.92 and 0.92, respectively. According to the scale developers, when participants report four out of eight problematic CPU symptoms and at least one functional impairment, they were categorized into the cellarer-phone-addiction group for screening and analysis purposes.

#### 2.2.3. ADHD Symptoms

The short version of the SNAP-IV (Chinese version) was used to assess the severity of ADHD symptoms exhibited in the preceding month. This version comprises 26-items encompassing the core DSM-4-derived ADHD subscales of inattention, hyperactivity and impulsivity, and the oppositional symptoms of oppositional defiant disorder [27,28]. Each item is rated on a 4-point Likert scale ranging from 0 (not at all) to 3 (very much). The Cronbach’s α for inattention, hyperactivity and impulsivity, and oppositional behavior were 0.91, 0.90, and 0.93, respectively, indicating very high internal consistency.

### 2.3. Analysis

The PSUMS was administered to the 211 study participants to determine its psychometric characteristics. Construct validity was determined by using factor analysis. Specifically, 211 participants were randomly divided into two subsamples using the RANDBETWEEN function in EXCEL. The first subsample (*n* = 103) was then used for exploratory factor analysis (EFA) and the second sample (*n* = 108) was used for confirmatory factor analysis (CFA). The EFA was conducted using SPSS 24 statistical software (SPSS Inc., Chicago, IL, USA). Based on the assumption that the factors are correlated, principal axis analysis was carried out with Promax rotation, after which the Kaiser–Mayer–Olkin (KMO) measure of sampling adequacy and Bartlett testing were applied. A KMO value of >0.60 and significant statistics from Bartlett testing suggested the data was suitable for factor analysis [31]. The amount of variance accounted for indicates how well a relevant notion or construct can be measured [32].

Since the multivariate skewness (4929.18) and multivariate kurtosis (40.25) indicate the non-normal distribution for the CFA subsample, maximum likelihood with Satorra-Bentler correction was performed to determine whether the model data fit with the item-factor structures obtained from the EFA. The CFA was conducted using LAVAAN package in the R software [33]. In general, the results indicate a good fit when the standardized root mean squared residual (SRMR) below 0.08 [34,35], root mean square error of approximation (RMSEA) below 0.08 [36,37], comparative fit index (CFI) and non-normed fit index (NNFI) above 0.9 [38,39], and a normed chi-square (i.e., chi-square/df) less than 2 [31] or less than 5 [40].

Two approaches for testing criterion-related validity were applied: known-group validity and concurrent validity. Known-group validity was determined by performing independent sample *t* test between the smartphone-addiction group and non-smartphone-addiction group. The concurrent validity of the PSUMS was determined by examining the correlation between the PSUMS and the PCPU-Q. Internal consistency was tested to assess the reliability of the scale.

## 3. Results

The preliminary analysis indicated that none of the study variables were skewed or kurtotic. The sociodemographic characteristics of parents and adolescents, adolescents’ ADHD symptoms, and smartphone-addiction symptoms are presented in Table 1. Descriptive statistics along with both the EFA and CFA factor loadings of the PSUMS items are presented in Table 2.

### 3.1. Construct Validity

#### 3.1.1. Exploratory Factor Analysis (EFA)

An EFA was conducted on a sample of 211 parents of children with ADHD. The KMO coefficient of sampling adequacy was 0.90 which lies within the excellent range. Bartlett’s Test of Sphericity, which assesses whether a matrix differs from the identity matrix, yielded significant results, indicating that the matrix did not resemble the identity matrix; this also supported the presence of factors within the data. Principal axis factor analysis was conducted and the Promax rotation method, which assumes factors are correlated and rotates the factor structure, was used to determine the factor solutions. The initial result confirmed the proposed three-factor solution for all 20 items. After eliminating three items with cross-loadings, a factor analysis was run once more on the remaining 17 items. The results supported a three-factor solution that explained 78.58% of the total variance. The items and factor loadings for the subscales are presented in Table 2. The three-factor solution fits with the theoretical factors used to devise the measurement tool. The first factor, “reactive management” (*α* = 0.93) includes seven items with factor loadings of 0.52–0.73; the items reflect parents’ reactive management of children’s smartphone use through rule-setting practices, responding to and controlling this use to avoid negative impacts on children’s daily-life functioning. The second subscale, “proactive management” (*α* = 0.95) includes 6 items with factor loadings of 0.55–0.93; the items reflect parents’ perceived efficacy of their proactive management and active mediation of children’s smartphone use through positive communication and reasoning. The third factor, “monitoring” (*α* = 0.93) includes 4 items with factor loadings of 0.63–0.91; the items reflect parents’ behavior in monitoring what their children do on their smartphones, whom they talk with, what applications they use, and the websites they visit. The score for each subscale was calculated by its mean. Pearson’s correlations among the dimensions of the PSUMS are presented in Table 3.

#### 3.1.2. Confirmatory Factor Analysis

The CFA fit indices were acceptable: *CFI* = 0.934, *NNFI* = 0.923, *RMSEA* = 0.077, and *SRMR* = 0.053, normed chi-square = 1.64; except for the significant chi-square test (chi-square = 189.68, *df* = 116; *p* < 0.001). Moreover, the factor loadings were strong in each factor: loadings = 0.80 to 0.90 in Reactive Management; 0.82 to 0.93 in Proactive Management; and 0.85 to 0.96 in Monitoring. Overall, the current model showed an acceptable fit to the data.

### 3.2. Criterion Validity

The PCPU-Q was used to determine the criterion-related validity of the PSUMS. After checking the equality of the variance, independent sample *t* test results provided strong evidence for known-groups validity, a subtype of criterion-related validity. We compared the PSUMS results between the parents of children with and without smartphone addiction. The results showed that parents of children with smartphone addiction yielded lower scores on all three PSUMS subscales than parents of children without smartphone addiction (Table 4). The correlation coefficients between three subscales of the PSUMS and two subscales of the PCPU-Q were also calculated. The results of the Pearson’s correlation provided strong evidence for concurrent validity (Table 5). The PSUMS was found to be significantly correlated with the PCPU-Q (all *p* < 0.001).

### 3.3. Reliability

The internal consistency coefficient (Cronbach’s *α*) of the PSUMS was 0.96. As shown in Table 2, the internal consistency coefficients in the sub-dimensions of the PSUMS were 0.93 for Reactive management, 0.95 for Proactive management, and 0.93 for Monitoring.

## 4. Discussion

In this study we developed and validated the PSUMS to measure parents’ perceived efficacy in managing the smartphone use of adolescents with ADHD. The final version of the 17-item PSUMS accounted for 78.58% of the total variance and contains three theoretically and statistically appropriate subscales: reactive management, proactive management, and monitoring. The three-factor hypothesis was supported by the scale’s psychometric properties, including its construct validity (confirmed using EFA and CFA), criterion-related validity (known-group validity and concurrent validity), and reliability (internal consistency measured using Cronbach’s alpha). The PSUMS adhered to psychometric standards and was shown to be a concise and promising measure of whether parents of adolescents with ADHD have the knowledge and skills required to successfully manage their children’s smartphone use.

The PSUMS consists of three scales. With a rating scale from 0 to 6, an increase in scores corresponds to an increase in parents’ efficacy in managing their children’s smartphone use. The three smartphone-management items parents feel most confident about were in the sub-dimension “reactive management”. These were: “I manage how and to what extent my child spends money on his/her smartphone”, “I don’t allow my child to use a smartphone while doing homework”, and “when my child is spending too much time on a smartphone, I stop his/her smartphone use effectively”. The two smartphone-management items they felt least confident about were in the sub-dimension “proactive management,” which involved positive communication and reasoning. These were: “I don’t get angry when I manage my child’s smartphone use”, and “I don’t distress my child when communicating with him/her about smartphone use”. Moreover, parents also produced low scores in regard to managing children’s smartphone use outside of the house.

In terms of criterion-related validity, the subscales of the PSUMS were significantly correlated in the expected direction with the PCPU-Q. The first subscale, reactive management, concerns parental intervention in adolescents’ smartphone overuse, money spending, cybercrime involvement, poor timing of use, and other smartphone behaviors that may negatively affect their daily lives. Adolescents whose parents use moderate behavioral control reported fewer problem behaviors [41,42]. Similarly, parents who set rules on media time reported that their children (aged 0–6 years) watched less television on average than other children in their age group [43]. As expected, reactive management was negatively correlated with smartphone addiction and functional impairments. Parents who scored lower on reactive management were more likely to have children with a smartphone addiction and functional impairments. The second subscale, proactive management, addresses parents’ positive communication, affection toward, and reasoning with adolescents about their smartphone use. Research has shown that adolescents who exhibit the symptoms and characteristics of ADHD are more likely than other children to experience relatively high levels of parent–child conflict [25]. In our study, effective communication, high levels of affection, and reasoning around smartphone use behaviors with adolescents with ADHD was found to be a key element in the successful prevention of problematic smartphone use. Proactive management enables adolescents to discuss their feelings, negotiate rules, and communicate in a safe and caring environment. This type of parenting practice has been linked to prosocial behavior, less externalizing behavior, good mental health, and higher achievement [44,45,46,47]. As expected, proactive management was negatively correlated with smartphone addiction and functional impairments. Finally, the third subscale, parental monitoring, was defined as “a set of correlated parenting behaviors involving attention to and tracking of the child’s whereabouts, activities, and adaptations” on their smartphone [48]. Parental monitoring has been associated with lower levels of alcohol and substance use [49,50]. Similarly, lower levels of parental monitoring have been found to correlate with a higher percentage of time spent on nonacademic computer use [51]. In this study, parental monitoring was negatively associated with smartphone addiction and functional impairment. The PSUMS was significantly correlated in the expected direction with the PCPU-Q, which confirmed its concurrent validity.

The PSUMS conceptualized and measured parents’ self-efficacy regarding their management of the smartphone use of adolescents with ADHD. However, this research had some limitations that need to be addressed. The sample sizes used for EFA and CFA were relatively small (i.e., 103 for EFA and 108 for CFA, total of 211). However, we believed that our contribution adds on the current literature because it is hard to collect more than a hundred adolescents with diagnosed ADHD. Furthermore, studies have shown that our sample sizes were acceptable for both EFA and CFA. For EFA, a rule of thumb to calculate the sample size is using an item-participant ratio of 5 [52]. Given that the original item number of the PSUMS is 20, 100 was a sufficient number for conducting EFA. Regarding CFA, although the consensus is that 200 is a preferable sample size, Iacobucci argue that sample size at 50 is sufficient [53]. Anderson and Gerbing [54] and Kline [55] agree that using a sample size of 100 could be the minimum criterion for doing CFA. Therefore, we tentatively concluded that our sample sizes were acceptable in both EFA and CFA [56]. Nevertheless, future studies are warranted to collect a larger sample size to corroborate our findings. Although construct validity, known-groups validity and concurrent validity were supported by the results, another limitation is that all subscales of PSUMS are significantly correlated with all subscales of PCPU-Q (the criterion scale), which indicated that subscales of PSUMS may not be distinguished. However, when performing the calculation for the test of the differences between two dependent correlations with one variable in common [57], we found a significant difference between two correlations of PSUMS subscales (reactive management and monitoring) with one subscale of PCPU-Q (functional impairment) in common as criterion (*z* = 2.49, *p* < 0.01). This limitation maybe because of the nature of smartphone addiction (PCPU-Q) is related to parental smartphone use management (PSUMS), thus subscales of PSUMS cannot be fully distinguished. Future studies may use monitoring scales as criterion in testing discriminant validity in PSUMS. Despite these limitations, the overall results showed that the PSUMS has good reliability and adequate validity. The reliability and validity of the scale mean it can be used to test different samples. Furthermore, the PSUMS could also be used as a tool for developing smartphone-addiction prevention programs for adolescents.

## 5. Conclusions

The PSUMS is considered a valuable and reliable tool in the study of parental management on their adolescent children’s smartphone use, while providing us with important targets for intervention. This study provides a foundation for future research in education, family science, and technology science by examining the multidimensional factors associated with parents’ efficacy in managing adolescents’ smartphone use in the digital age. Because Internet and smartphone are widely used and become an important part of our daily life, it is essential to enhance the traditional parenting and educational programs with consideration of the management on children/adolescents’ smartphone use, strength parents’ efficacy in doing so, and prepare parents with right tools to face this new challenge in parenting.

## Figures and Tables

**Table 1 ijerph-16-01423-t001:** Sociodemographic characteristics of parents and adolescents and the ADHD symptoms.

Variables	*n* (%)	Mean (*SD*)	Range
Sex of parents			
Female	175 (82.9%)		
Male	36 (17.1%)		
Sex of adolescents			
Female	28 (13.3%)		
Male	183 (86.7%)		
Age of parents (years)		43.5 (5.9)	32–64
Age of adolescents (years)		13.7 (1.8)	11–18
Marriage status of parents			
Intact	170 (80.6%)		
Not intact	41 (19.4%)		
Education duration of parents (years)		13.6 (2.8)	6–28
SNAP-IV symptoms of adolescents			
Inattention		12.8 (6.1)	0–27
Hyperactivity/impulsivity		8.9 (6.0)	0–27
Oppositional defiant		9.9 (5.7)	0–24
PCPU-Q			
Smartphone addiction	40 (19%)		
No smartphone addiction	171 (81%)		

Note: SNAP-IV: Short version of the Swanson, Nolan, and Pelham Version IV Scale—Chinese version. PCPU-Q: Problematic Cellular Phone Use Questionnaire. *SD* = standard deviation.

**Table 2 ijerph-16-01423-t002:** Means, standard deviations, Cronbach’s α, and factor loadings for the items in the PSUMS.

Items	Mean (*SD*) (*n* = 211)	EFA *n* = 103	CFA *n* = 108
Reactive Management (*α* = 0.93)			
I manage my child’s smartphone use to prevent it from negatively affecting his/her daily life	4.35 (1.43)	0.73	0.89
I manage how and to what extent my child spends money on his/her smartphone	4.64 (1.44)	0.66	0.81
I don’t allow my child to use a smartphone while doing homework	4.42 (1.48)	0.59	0.86
I manage my child’s smartphone use outside of the house	3.74 (1.76)	0.59	0.80
I effectively manage when my child can and cannot use a smartphone	4.18 (1.61)	0.57	0.88
I manage my child’s activities to prevent him/her from breaking laws	4.30 (1.49)	0.53	0.83
When my child is spending too much time on a smartphone, I manage his/her smartphone use effectively	4.38 (1.40)	0.52	0.90
Proactive Management (*α* = 0.95)			
I don’t distress my child when communicating with him/her about smartphone use	3.72 (1.58)	0.93	0.91
I don’t create family tension as a result of enforcing smartphone use guidelines for my child	3.81 (1.55)	0.82	0.93
I discuss and reason with my child	4.11 (1.37)	0.80	0.93
I don’t get angry when I manage my child’s smartphone use	3.53 (1.61)	0.72	0.83
I communicate with my child effectively and explain why I manage his/her smartphone use	4.22 (1.33)	0.70	0.93
I actively learn new information and skills to manage my child’s smartphone use	4.14 (1.44)	0.55	0.82
Monitoring (*α* = 0.93)			
I know who my child talks with and what they talk about when using a smartphone	3.81 (1.69)	0.91	0.96
I know what my child does on the smartphone	3.86 (1.64)	0.84	0.95
I monitor which apps my child uses	3.79 (1.69)	0.72	0.88
I restrict the type of websites my child is allowed to visit on the smartphone	3.91 (1.69)	0.63	0.85

Note: EFA = exploratory factor analysis; CFA = confirmatory factor analysis; *SD* = standard deviation.

**Table 3 ijerph-16-01423-t003:** Correlation among the dimensions of the PSUMS.

PSUMS Dimensions	Reactive Management	Proactive Management	Monitoring
Reactive management	-		
Proactive management	0.79 **	-	
Monitoring	0.77 **	0.68 **	-

Note: ** *p* < 0.01.

**Table 4 ijerph-16-01423-t004:** Parental smartphone use management among children with and without smartphone addiction.

PSUMS Dimensions	Have Smartphone Addiction (*n* = 40) Mean (*SD*)	No Smartphone Addiction (*n* = 171) Mean (*SD*)	*t*	*p*
Reactive management	3.30 (1.46)	4.52 (1.14)	4.93	<0.001
Proactive management	2.90 (1.39)	4.11 (1.27)	5.34	<0.001
Monitoring	2.94 (1.56)	4.05 (1.47)	4.25	<0.001

Note: *SD*: Standard Deviation.

**Table 5 ijerph-16-01423-t005:** Correlation between Parental Smartphone Use Management Scale (PSUMS) and PCPU-Q.

PCPU-Q Dimensions	Parental Smartphone Use Management (PSUMS)		
	Reactive Management	Proactive Management	Monitoring
	*r*	*r*	*r*
Smartphone addiction symptoms	−0.35 **	−0.40 **	−0.33 **
Functional impairments	−0.37 **	−0.33 **	−0.26 **

Note: ** *p* < 0.01. PCPU-Q: Problematic Cellular Phone Use Questionnaire.

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
