# Peer review of "Development and Validation of the Parental Smartphone Use Management Scale (PSUMS): Parents’ Perceived Self-Efficacy with Adolescents with Attention Deficit Hyperactivity Disorder"

_ijerph, 2019, doi:10.3390/ijerph16081423_

Round 1

Reviewer 1 Report

The authors developed a tool for measuring parental smartphone use management (PSUMS). They provide various results to support the validity of PSUMS. Because the smartphone overuse and addiction in adolescents are a big social issue, I believe that this work is valuable. However, there are some concerns as below. 

1. In this paper, the authors used only 211 samples with ADHD. Is this sample size enough to expect similar results in the other sample?

2. The authors applied both EFA and CFA on the same dataset. When using the same data, a factor model derived from an EFA is usually fit well in a CFA. So, it is no surprise that the model fit indices are good (Section 3.1.2). I recommend that CFA should be applied in the independent sample.

3. In table 4, the authors only showed Pearson’s correlation between the subscales of PSUMS and PCPU-Q. However, all subscales are significantly correlated with all subscales of PCPU-Q. In this case, it can be understood that subscales of PSUMS cannot be distinguished. Clear interpretation and explanation are necessary.

Author Response

1. In this paper, the authors used only 211 samples with ADHD. Is this sample size enough to expect similar results in the other sample?

 [Response]: We added the following as a limitation into the Discussion Section. The sample sizes used for EFA and CFA were relatively small (i.e., 103 for EFA and 108 for CFA and 211 in total). However, we believed that our contribution adds on the current literature because it is hard to collect more than a hundred adolescents with diagnosed ADHD. Moreover, studies have shown that our sample sizes were acceptable for both EFA and CFA. For EFA, a rule of thumb to calculate the sample size is using item-participant ratio at 5 [52]. Given that the original item number of the PSUMS is 20, 100 was a sufficient number for conducting EFA. Regarding CFA, although the consensus is that 200 is a preferable sample size, Iacobucci argue that sample size at 50 is sufficient [53]. Moreover, Anderson and Gerbing [54] and Kline [55] both agree that using sample size at 100 could be the minimum criterion for doing CFA. Therefore, we tentatively concluded that our sample sizes were acceptable in both EFA and CFA [56]. Nevertheless, future studies are warranted to collect a large sample size to corroborate our findings.

2. The authors applied both EFA and CFA on the same dataset. When using the same data, a factor model derived from an EFA is usually fit well in a CFA. So, it is no surprise that the model fit indices are good (Section 3.1.2). I recommend that CFA should be applied in the independent sample.

[Response]: Thank you for the suggestion. We re-run the EFA and CFA using two independent samples. The 211 participants were randomly divided into two subsamples using the randbetween function in the EXCEL. The first subsample (n=103) was then used for exploratory factor analysis (EFA) and the second sample (n=108) was used for confirmatory factor analysis (CFA).

3. In table 4, the authors only showed Pearson’s correlation between the subscales of PSUMS and PCPU-Q. However, all subscales are significantly correlated with all subscales of PCPU-Q. In this case, it can be understood that subscales of PSUMS cannot be distinguished. Clear interpretation and explanation are necessary.

[Response]: We discuss it as a limitation that all subscales of PSUMS are significantly correlated with all subscales of PCPU-Q (the criterion scale). However, when performed calculation for the test of the difference between two dependent correlations with one variable in common [57], we found the significant difference between two correlations of PSUMS subscales (reactive management and monitoring) with one subscale of PCPU-Q (functional impairment) in common as criterion (z = 2.49, p < .01). This limitation may be because of the nature of smartphone addiction (PCPU-Q) is related to parental smartphone use management (PSUMS), thus subscales of PSUMS cannot be fully distinguished. Future studies may use monitoring scales as criterion in testing discriminant validity in PSUMS.

Reviewer 2 Report

The paper discusses an interesting mater. However, it requires some improvements. The introduction covers the main topic of the problem.

Analysis. Please report here the cut-offs of the CFA fit indices. In the next section, please report only the results of the CFA. In this section, can you please provide information concerning the multivariate skewness and kurtosis? It is important to discuss this issue because of the higher value of RMSEA. In addition, please report the EFA results as well as the chi-square statistic (chi-square/DF ratio), which a rule of thumb is < 2.0 or less conservatively < 5.0. Please report also the correlation among the dimensions of the PSUMS.

Author Response

The paper discusses an interesting mater. However, it requires some improvements. The introduction covers the main topic of the problem.

Analysis. Please report here the cut-offs of the CFA fit indices. In the next section, please report only the results of the CFA. In this section, can you please provide information concerning the multivariate skewness and kurtosis? It is important to discuss this issue because of the higher value of RMSEA. In addition, please report the EFA results as well as the chi-square statistic (chi-square/DF ratio), which a rule of thumb is < 2.0 or less conservatively < 5.0. Please report also the correlation among the dimensions of the PSUMS.

[Response]: Thank you for the comments. (1) The cut-offs of the CFA fit indices have been reported in 2.3. Analysis. (2) We added information of the multivariate skewness and kurtosis in 2.3. Analysis. (3) The result of chi-square statistic (chi-square/DF ratio) has been reported in 3.1.2. Confirmatory Factor Analysis. (4) The correlation among the dimensions of the PSUMS has been reported in Table 3.

Round 2

Reviewer 1 Report

The revision is generally satisfactory.

Author Response

I think there are still minor adjustments to do:
- correcting typos
- introducing acronyms first (even statistical ones)
- italics in all statistics
- references correctly managed

[Response]: Thanks for the comments. These adjustments were made.

Another suggestion to clarify the work they are presenting (a scale for parents to measure their perceived self-efficacy on their children's smartphone use) 
I would like to ask you please to tell to the authors to add important data about the work in the title and some sections.
Title (provisional suggestion):  Development and Validation of the Parental Smartphone Use Management Scale (PSUMS): perceived self-efficacy of parents with children with attention deficit hyperactivity disorder
[Response]: Thanks for the suggestion. The title has been changed to “Development and Validation of the Parental Smartphone Use Management Scale (PSUMS): Parents’ Perceived Self-Efficacy with Adolescents with Attention Deficit Hyperactivity Disorder.”

Similarly, in the (final) Introduction and (beginning) method sections (my suggestions are between brackets):
"Therefore, a reliable metric is needed to assess (self-perceived) parents’ self-efficacy regarding management of their (ADHD) adolescent children’s smartphone use"
[Response]: The information was added.

"In response to their opinions and criticism, the scale was revised to remove inappropriate
wording and irrelevant content (e.g., XXX). Parents were then asked to rate their (self-perceived) efficacy in managing their children’s smartphone use over the previous month. High PSUMS scores indicate high levels of self-efficacy."
[Response]: The word was added and the sentence was modified. “…the scale was revised to use culture-sensitive wording and remove irrelevant content.”